# Turn-On Fluorescence Aptasensor on Magnetic Nanobeads for Aflatoxin M1 Detection Based on an Exonuclease III-Assisted Signal Amplification Strategy

**DOI:** 10.3390/nano9010104

**Published:** 2019-01-16

**Authors:** Fuyuan Zhang, Linyang Liu, Shengnan Ni, Jiankang Deng, Guo-Jun Liu, Ryan Middleton, David W. Inglis, Shuo Wang, Guozhen Liu

**Affiliations:** 1ARC Centre of Excellence in Nanoscale Biophotonics (CNBP), Macquarie University, North Ryde, NSW 2109, Australia; fuyuan.zhang@aliyun.com (F.Z.); david.inglis@mq.edu.au (D.W.I.); 2Graduate School of Biomedical Engineering, Faculty of Engineering, University of New South Wales, Sydney, NSW 2052, Australia; linyang.liu@unsw.edu.au; 3State Key Laboratory of Food Nutrition and Safety, Key Laboratory of Food Nutrition and Safety, Ministry of Education of China, Tianjin University of Science and Technology, Tianjin 300457, China; dengjk1989@163.com (J.D.); s.wang@tust.edu.cn (S.W.); 4International Joint Research Center for Intelligent Biosensor Technology and Health, College of Chemistry, Central China Normal University, Wuhan 430079, China; 15527604959@163.com; 5Australian Nuclear Science and Technology Organization, Lucas Heights, NSW 2234, Australia; gdl@ansto.gov.au (G.-J.L.); rym@ansto.gov.au (R.M.); 6Discipline of Medical Imaging & Radiation Sciences, Faculty of Medicine and Health, Brain and Mind Centre, University of Sydney, 94 Mallett Street, Camperdown, NSW 2050, Australia; 7Tianjin Key Laboratory of Food Science and Health, School of Medicine, Nankai University, Tianjin 300071, China

**Keywords:** aflatoxin M1, magnetic nanobeads, aptasensors, G-quadruplex, signal amplification

## Abstract

In order to satisfy the need for sensitive detection of Aflatoxin M1 (AFM1), we constructed a simple and signal-on fluorescence aptasensor based on an autocatalytic Exonuclease III (Exo III)-assisted signal amplification strategy. In this sensor, the DNA hybridization on magnetic nanobeads could be triggered by the target AFM1, resulting in the release of a single-stranded DNA to induce an Exo III-assisted signal amplification, in which numerous G-quadruplex structures would be produced and then associated with the fluorescent dye to generate significantly amplified fluorescence signals resulting in the increased sensitivity. Under the optimized conditions, this aptasensor was able to detect AFM1 with a practical detection limit of 9.73 ng kg^−1^ in milk samples. Furthermore, the prepared sensor was successfully used for detection of AFM1 in the commercially available milk samples with the recovery percentages ranging from 80.13% to 108.67%. Also, the sensor performance was evaluated by the commercial immunoassay kit with satisfactory results.

## 1. Introduction

Milk and dairy products are consumed in large amounts by people at different ages worldwide. Consequently, the safety and quality of milk are of immense importance for human health. However, once lactating mammals have consumed animal feed contaminated with aflatoxin B1 (AFB1), a highly toxic mycotoxin that is extensively present in crops, a metabolite of aflatoxin M1 (AFM1) will be produced in the liver through hydroxylation under the cytochrome P450 catalysis and then subsequently secreted in the milk [1,2,3,4]. Many surveys from different regions and countries revealed that AFM1 was widely observed in milk and milk products of those lactating cows exposed to AFB1 contaminated feed [5,6,7,8]. In humans, ingestion of the AFM1 contaminated milk and dairy products may cause both acute and chronic toxicoses, for example, hepatotoxicity, carcinogenicity, cytotoxicity, mutagenicity and genotoxicity [1]. Considering the health risks, AFM1 was categorized as a group 1 human carcinogen by the International Agency for Research on Cancer (IARC) [9,10] and many regions and countries have set legal limits for maximum residue level (MRL) of AFM1 in milk. For example, the United States’ Food and Drug Administration (FDA) has set a stringent MRL value of 500 ng kg^−1^ in milk. The MRL set by the European Union is 50 ng kg^−1^ for milk and milk-based products and it is much lower for infant milk and baby food (25 ng kg^−1^) [11,12]. Thus, the sensitivity is the long-standing issue for detection of AFM1.

Usually, instrumental analytical methods such as high-performance liquid chromatography (HPLC) [13], thin layer chromatography (TLC) [14] and liquid chromatography coupled with mass spectroscopy (LC-MS) [15] are conducted to monitor the AFM1 but the requirement of well-equipped laboratories and facilities restricts the wide application of these methods. Conversely, various immunoassays like enzyme-linked immunosorbent assays (ELISA) [16], fluoroimmunoassays (FLISA) [17] and immunosensors [18] have been developed for AFM1 detection based on the antigen-antibody specific recognition. Although the immunoassays have the advantages of high sensitivity, relative rapidity and high throughput, preparation of an antibody with high specificity, sensitivity and batch-to-batch consistency is difficult and costly. Moreover, it is time-consuming to prepare a functional antibody through animal immunization and there also exist batch-to-batch differences [19]. Alternatively, an aptamer which is a short oligonucleotide sequence with a high affinity to the target, is increasingly being selected and used as the recognition molecule in analytical chemistry and food safety [20]. It has the advantages of low cost, high stability, easy synthesis and flexible modification compared with antibody [21].

Many kinds of AFM1 aptasensors have been constructed ever since the aptamers with high affinity against AFM1 were screened [22,23,24,25]. Atul et al. developed a structure-switching signalling aptamer assay based on the FAM-TAMRA quenching-dequenching mechanism and a detection limit of 5.0 ng kg^−1^ for AFM1 has reached [11]. Noguer et al. designed an AFM1 electrochemical biosensor with a detection limit of 1.15 ng L^−1^ by immobilizing the AFM1 aptamer on the carbon screen-printed electrode [12]. However, most of these strategies suffered from high background signals from dye-aptamer non-specific binding and thus sensitivity and specificity have been sacrificed. To meet the demands of the low detection limit of AFM1, different signal amplification approaches have been introduced into the DNA-based detection methodologies in the most recent studies, such as the enzyme-assisted target recycling and DNA replication machinery [26]. Yin et al. constructed a label-free signal amplification aptasensor for AFM1 detection in milk, which showed a limit of detection of 19.7 ng kg^−1^ [27]. Although the developed aptasensors obtained lower detection limits by introducing signal amplification process, the designed amplification approaches were complex and time-consuming since they were involved with sophisticated steps such as applying several different enzymes [27].

To address this challenge, we incorporated a simpler signal-on fluorescence aptasensor for ultrasensitive detection of AFM1 on the basis of an autocatalytic Exonuclease III (Exo III)-assisted signal amplification (Figure 1). In addition, we also took advantages of nanomaterials (magnetic nanobeads) such as signal amplification and easy separation. In this designed sensor, with the presence of AFM1, a single-stranded DNA (C-strand) will be released from a DNA nanomachine which is formed by a duplex DNA (aptamer/C) and magnetic nanobeads. This released C-strand can trigger a simple fluorescence signal amplification recycle. Exo III is used as an amplifying biocatalyst in this amplification procedure to selectively digest duplex DNAs from blunt or recessed 3’-termini to produce a guanine-rich DNA strand with a G-quadruplex structure [28,29]. In the absence of C-strand, the 3’ protruding duplex DNA formed by G-strand and T-strand cannot be digested and there is no G-quadruplex structure produced in the system. The signal amplification process is label-free without any covalent labelling of luminophores but instead the N-methyl mesoporphyrin IX (NMM) is attached to the G-quadruplex DNA through weak interactions. NMM is an organic fluorescence dye which is only slightly emissive in aqueous solution, resulting from quenching of the excited state by solvent interactions [30,31]. However, it can emit significantly amplified fluorescence signals upon binding to the G-quadruplex structure owing to the protection of their excited states within the hydrophobic interior of the oligonucleotide [26,32]. This signal amplification strategy just consists of one enzyme actuated one-step recycle process, and the label-free strategy can overcome the reduction of the recognition event caused by the covalent attachment and reduce the assay time.

In addition to the novelty of the simple one-step label-free fluorescence signal amplification recycle, magnetic nanobeads were introduced in our assay which provide the big surface area to load the duplex DNA. Additionally, magnetic nanobeads are helpful to remove the uncaptured aptamer strands remained in the first-step resultant solutions. This can increase the stability of the assay by avoiding the aptamer strands competitively bind to the C-strand in the signal amplification recycle. Moreover, the magnetic nanobeads served as a reaction platform which can be removed together with the residual aptamer/C duplex and aptamer strand in a magnetic field after the first-step reaction. Consequently, due to the high affinity of aptamers to AFMI, AFM1 can trigger the signal-on amplification strategy, resulting in an enhanced fluorescence of NMM. This simple indirect sensing strategy was successfully applied to the sensitive monitoring of AFM1 in whole milk, skimmed milk and juice milk with the sensitivity of 0.01 ng mL^−1^ in milk samples.

## 2. Experimental

### 2.1. Reagents and Apparatus

Functional magnetic nanobeads (500 nm) were obtained from Ocean Nanotech (San Diego, CA, USA). All the used DNA strands and aptamers were synthesized and purified using HPLC by Sangon Biotech Co. (Shanghai, China) (Appendix A). MgCl_2_, KCl, tris (hydroxymethyl) aminomethane (Tris), *N*-hydroxysulfosuccinimide (NHS), 1-ethyl-3-(3-dimethylaminopropyl) carbodiimide (EDC), streptavidin and chemical standards of AFM1, AFM2, AFB1 and Ochratoxin A were purchased from Sigma-Aldrich (St. Louis, MO, USA). N-methyl mesoporphyrin IX (NMM) was obtained from Frontier Scientific (Logan, UT, USA). Exo III was purchased from Genesearch Pty Ltd. (Arundel, QLD, Australia). Milli-Q water was prepared using a Millipore water system (Billerica, MA, USA). Whole milk, skimmed milk and juice milk samples were commercially purchased from the local market. The Tris buffer (10 mM Tris-HCl, 10 mM MgCl_2_, 75 mM KCl, pH 8.0) was used in the experiment.

NanoDrop purchased from Thermo-Fisher Scientific (Waltham, MA, USA) was applied to redefine the concentration of the used DNA strands and aptamers. A metal bath obtained from Thermo-Fisher Scientific served to control the temperature of the recycle amplification. The fluorescence intensities at 612 nm were recorded using the Cary Eclipse fluorescence spectrophotometer from Agilent Technologies (Varian, CA, USA). A commercial ELISA kit for AFM1 was purchased from Sangon Biotech Co., Ltd. (Shanghai, China) to verify the established aptasensors.

### 2.2. Assembly of the Proposed Aptasensor

0.5 µM of AFM1 aptamer and 1 mg mL^−1^ of streptavidin immobilized magnetic nanobeads were sufficiently mixed together in Tris buffer (pH 7.4), followed by incubation at 37 °C for 30 min. The mixture was then washed three times using Tris buffer in a magnetic field to remove the uncombined aptamer. After that, the complementary single-stranded DNA (C-strand DNA in Tris buffer, 0.5 µM) was added to the above nanobeads and sufficiently mixed at 37 °C for 30 min to form the double-stranded DNA (aptamer/C duplex) with the aptamer strands-immobilized nanobeads. The resultant solution was subsequently washed three times using Tris buffer in the magnetic field to remove the uncombined C-strand DNA fragments. Simultaneously, the same volume of single-stranded template DNA (T-strand DNA, 1 µM) and guanine-rich DNA (G-strand, 1 µM) in Tris buffer were pre-heated to 90 °C for 3 min, followed by cooling slowly to 37 °C and incubated for 1 h to facilitate the hybridization of both strands and form another double strand DNA (G/T duplex) which would be used in the amplification recycle.

### 2.3. Monitoring of the AFM1 Based on the Proposed Aptasensor

Samples or a series of AFM1 standards with different concentrations (50 µL) were incubated with 50 µL of the aptamer/C duplex-coated magnetic nanobeads for 40 min at 37 °C to let the target AFM1 bind to the aptamer and then replaced the C-stand DNA. Subsequently, the supernatant which contained the released C-strand DNA was separated carefully by the magnetic field for 5 min. Then 5 µL of the resultant supernatant was added to 45 µL of Tris buffer solution which contained 0.5 µM of G/T duplex and 20 U of Exo III. Next, the mixture started a process of hybridization at 37 °C for 40 min followed by the treatment at 72 °C lasting for 20 min. Lastly, NMM, with a final concentration of 1 µM, was added to the system and incubated for 15 min. The subsequent products were submitted to the Cary Eclipse fluorescence spectrophotometer to record the fluorescence intensity signals.

### 2.4. The Pre-Treatment of Milk Samples

Three kinds of milk and milk products (whole milk, skimmed milk and juice milk) were introduced into the established aptasensors to perform recovery experiments. Before being applied in the aptasensor, all used samples were pre-detected by a commercial ELISA kit and no positive results were collected. Specifically, samples were shaken well before the treatment. Subsequently, the upper creamy layer was removed through a centrifugation process (4 °C, 10 min, 5000 rpm) and the remaining liquid was collected after the centrifugation process. Then, a series of AFM1 standards was spiked to the milk remaining liquid with the stirring for 20 min. The milk samples with a different concentration of AFM1 (20, 50 and 100 ng kg^−1^) were used in the assay. A similar sample preparation procedure was adopted for the analysis of AFM1 using the commercial ELISA Kit.

## 3. Results and Discussions

### 3.1. Design of the Proposed Sensing System

The proposed protocol includes two functional parts: one is an AFM1 induced strand release process and the other part is a signal amplification recycle based on a strand displacement (Figure 1). The AFM1 induced strand release process involved an AFM1 triggered DNA nanomachine which was assembled by aptamer/C duplex DNA and magnetic nanobeads. At first, a high-affinity anti-AFM1 aptamer was modified with biotin and captured by a streptavidin immobilized magnetic nanobead through highly specific biotin-streptavidin interaction. The captured aptamer hybridized with its complementary strand (C-strand) to form a duplex DNA (aptamer/C duplex). AFM1 is able to dissociate the aptamer/C duplex owing to the specific affinity interaction between the aptamer and AFM1. Following washing in a magnetic field, the unopened aptamer/C duplexes were removed together with the magnetic nanomachine. The released C-strands were then submitted to the second part and functioned as the primer to trigger the signal amplification recycle.

In this recycle, a double strand DNA (G/T duplex) was pre-formed by adding a guanine-rich strand (G-strand) to hybridize with a template DNA (T-strand) which partly complemented the G-strand and C-strand, respectively. Thus, the C-strand DNA could recognize and hybridize with its complementary sequence in the T-strand through a toehold strand displacement, while forming a new duplex DNA containing a 3’-blunt terminus created by the C-strand, G-strand and T-strand. Under the function of the Exo III existing in the system, the 3’-blunt terminus of duplex DNA was selectively digested, resulting in the release of the G-strand and C-strand. The G-strand DNA followed to form a G-quadruplex structure with the help of K^+^ in the buffer and clearly played the role of a signal transducing probe to enhance the fluorescence signal of NMM. The released C-strand recognized and hybridized with the new G/T duplex DNA in the system again and started a new amplification recycle. As a result, the target AFM1 could produce a high fluorescence intensity of NMM by converting the AFM1 signal to the detection of C-stand through an AFM1 induced nanomachine and a signal amplification recycle triggered by C-strand.

### 3.2. Verification of the Designed Sensing System

To verify the feasibility of the Exo III-assistant signal amplification recycles in this aptasensor, the fluorescence signal of solutions containing different mixtures were recorded after incubation with NMM. As we can see in Figure 2, a fairly weak fluorescence signal was observed for the solution containing NMM alone because of the tiny fluorescence emission of NMM. Upon the addition of the G-strand, a 3.2-fold increase in the fluorescence intensity was obtained, owing to the link between NMM with the folded G-quadruplex structure of the G-strand. However, the introduction of the T-strand into the solution containing NMM alone simply generated a fluorescence signal as low as the background signal of NMM alone, due to the T-strand being unable to form a G-quadruplex structure to enhance the fluorescence intensity of the NMM. In addition, the mixture of NMM and G/T duplex DNA also exhibited an insignificant change in fluorescence intensity because the formation of G/T duplex led to the lack of guanine-rich sequence (G-strand).

Even though Exo III was further introduced into the above solution, any change in the fluorescence intensity was also negligible, because the toehold region on the G/T duplex was not digested by Exo III which was only specific to the 3’-blunt terminus on duplex DNA, resulting in none of the G-strand DNA being released. When the C-strand DNA was added to the solution containing the G/T duplex DNA and Exo III, the C-strand hybridized with the T-strand to form the 3’-blunt terminus on the duplex DNA. Subsequently, it was digested by Exo III, releasing the C-strand DNA for initiating another cycle and ultimately liberating much more G-stands to fold into the quadruplex structures to enhance the fluorescence intensity of NMM. Thus, a remarkable increase in fluorescence intensity (7.2-fold) was observed, which was much stronger than that caused by the G-strand alone. This result evidenced a signal amplification recycle in the solution containing Exo III, C-strand and G/T duplex DNA. It also revealed that the generation of the G-quadruplex structure highly depended on the C-strand and Exo III.

### 3.3. Binding Aptamers to the Magnetic Nanobeads

Besides the advantage of being easy washing in magnetic field, magnetic nanobeads serve as a nanocarrier for the recognition molecules (aptamers). Biotinylated aptamers are conjugated on the streptavidin-coated magnetic nanobeads via streptavidin-biotin interaction and the number of aptamers loaded on the nanobeads is critical to the sensitivity of the assay. In this study, the binding capacity of the magnetic nanobeads were tested (Figure 3a,b). Aptamers in the concentration of 0.90 ± 0.08 µM (*n* = 3) were captured by 1 mg mL^−1^ of magnetic nanobeads after a conjugation reaction. Subsequently, the resultant magnetic nanobeads were used to bind the C-strand DNA, which revealed that 720 ± 69 pmoL (*n* = 3) of the aptamer/C duplexes were formed on 1 mg of the magnetic nanobeads.

### 3.4. Optimization of Experimental Parameters

To introduce an efficient signal amplification strategy to this aptasensor for detection of AFM1, experimental conditions, such as the concentration of NMM, interaction time of the amplification and the AFM1 activated displacement reaction time were investigated. As depicted in Figure 3c, different concentrations of NMM were tested in the system containing 1 µM of G-strand DNA and fluorescence intensity reached its maximum value at the concentration of 1 µM and gradually decreased thereafter. It indicated that this system produced the highest fluorescence intensity when the molar ratio of formed G-quadruplex with NMM reached 1:1. Accordingly, the optimized concentration of NMM at 1 µM was applied in subsequent experiments.

Under the optimized amplification conditions, the AFM1 activated displacement reaction in the first step was optimized in Tris buffer. As illustrated in Figure 4a, different amounts of resultant solution in the first-step were submitted to the amplification recycle. When 10 µL and 20 µL of the resultant solution were tested, fluorescence changes did not promptly occur from 1 ng mL^−1^ to 20 ng mL^−1^ of AFM1. This was because the amount of C-strand in these two solutions was too high to induce distinguishable changes in fluorescence enhancement. Although there was an obvious signal enhancement trend when 1 µL and 2 µL of the solution were added, the highest fluorescence intensity was not high enough to make an assay effective. When 5 µL of the solution was submitted to the system, a significant effect was observed, suggesting that the highest fluorescence intensity was high enough to be utilized for measurement. Therefore, 5 µL of the solution was used in subsequent experiments. The AFM1 activated displacement reaction time optimization is shown in Figure 4b. The fluorescence response gradually increased along with increasing interaction time between AFM1 and aptamer/C duplex and then levelled off after 40 min, indicating that almost all target AFM1 was associated with aptamer strands and released the C-strand within 40 min. Accordingly, a reaction time of 40 min in the first step was used in this experiment.

The amplification recycle reaction time was optimized based on the above conditions. As shown in Figure 3d, the fluorescence response gradually increased along with increasing reaction time in the 10 to 40 min range and then levelled off following a longer incubation time. This revealed that an incubation time of 40 min was enough to complete the amplification recycle. Therefore, 40 min of amplification recycle reaction time was applied for the following experiments.

### 3.5. Performance of the Aptasensor

AFM1 with a series of concentrations was added to the negative control milk sample which was pre-treated according to the sample treatment method and used as the working buffer to establish the calibration curve of this aptasensor. Under the optimized conditions, the calibration curve was obtained by plotting between fluorescence intensity and concentrations. As we can see in Figure 4c,d, the fluorescence intensity increased when the AFM1 concentration increased from 0 ng mL^−1^ to 20 ng mL^−1^ and saturated when then concentration reached 10 ng mL^−1^. It confirmed that the AFM1 triggered the C-strand DNA which was released from the magnetic nanobeads and then induced the Exo III-assisted recycling amplification. Thus, the AFM1 signal was quantified by detection of the fluorescence signal of C-stand. We fitted the fluorescence signal gain to a dynamic four-parameter equation to obtain an apparent equilibrium dissociation constant (IC_50_) of 29.48 ng mL^-1^ with a regression coefficient of 0.993. The dynamic detection linear range of AFM1 was from 0.01 to 2 ng mL^−1^ with the correlation coefficient of 0.995. (Figure 4d). According to the practical detection results, 0.01 ng mL^−1^ was defined as the practical detection limit (equivalent to 9.73 ng kg^-1^), which has satisfied the most stringent MRL value of 25 ng kg^−1^ in infant milk and baby food by FDA. Comparison of the performance of this aptasensor with that of sensors reported in the literatures is listed in Appendix A. These results obviously proved the ability of the herein aptasensor to serve as a method for AFM1 sensing with high sensitivity due to the Exo III-assisted signal amplification.

### 3.6. Selectivity of the Proposed Aptasensor

The selectivity of this aptasensor was investigated by testing another three mycotoxins (AFB1, AFM2 and Ochratoxin A) which have a similar molecular structure to AFM1. Mycotoxins with a concentration of 1 ng mL^−1^ were tested, because this concentration was located in the middle of the detection linearity range. According to the results displayed in Figure 5, a significant signal enhancement was recorded for AFM1 compared to the blank milk sample. In contrast, the interference toxins (AFB1, AFM2 and Ochratoxin A) did not cause obvious changes in the fluorescence signal, indicating that the presence of these control mycotoxins showed an ignorable influence on this AFMI sensor. The reason lies in that the aptamer in the DNA nanomachine is specifically geared toward AFM1. Hence, the fabricated aptasensor has remarkable selectivity due to the affinity of aptamer to AFM1.

### 3.7. Analysis of AFM1 in Spiked Milk Samples

To demonstrate the capability of this aptasensor for detection of AFM1, the aptasensors were used for detection of AFM1 in three kinds of milk samples which were spiked in with AFM1 with the concentration of 20, 50 and 100 ng kg^−1^, respectively. The blank milk samples without AFM1 were detected firstly and no positive results were obtained. As shown in Figure 6 and Table 1, the average recovery of the sensor ranged from 80.13% to 108.67% in these four milk samples, suggesting this developed aptasensor was accurate and able to quantify AFM1 in real milk samples. The performance of this sensor was also verified by commercial ELISA kit. Results of sample analysis achieved by the aptasensor were in good agreement with those obtained from commercial ELISA kit (Table 1). The recovery ranged from 81.81% to 103.40% indicating the proposed sensor can be used for accurate and sensitive detection of AFM1 in milk samples. Compared with the ELISA method, this developed sensor has several advantages such as low cost, short assay time and simple operation steps.

## 4. Conclusions

In summary, we developed a AFM1-activated and DNA-fuelled fluorescence aptasensor for sensitive detection of AFM1 by employing an EXO III-based signal amplification strategy. This aptasensor has skilfully realised the detection of AFM1 by quantifying the fluorescence intensity of single strand DNA based on a target-induced strand displacement strategy. The released DNA strand can activate a simple EXO III-based signal amplification recycle to generate massive G-quadruplex structures for achieving significant signal enhancement for selective AFM1 detection down to 0.01 ng mL^−1^ in milk samples. In addition to increase the sensitivity by providing big surface area, the introduced magnetic nanobeads can separate the uncaptured strands effectively to control the sensoring stability. Besides, this aptasensor takes advantages of EXO III to digest the 3’-blunt terminus of duplex DNA and recycle the released strand, producing significantly amplified fluorescent signals. Moreover, the strategy studied herein is simple to design, easy to operate and holds great potential for constructing a wide range of fluorescently amplified aptasensors for detection a spectrum of analytes by optimizing aptamer/complementary strand combination.

## Figures and Tables

**Figure 1 nanomaterials-09-00104-f001:**
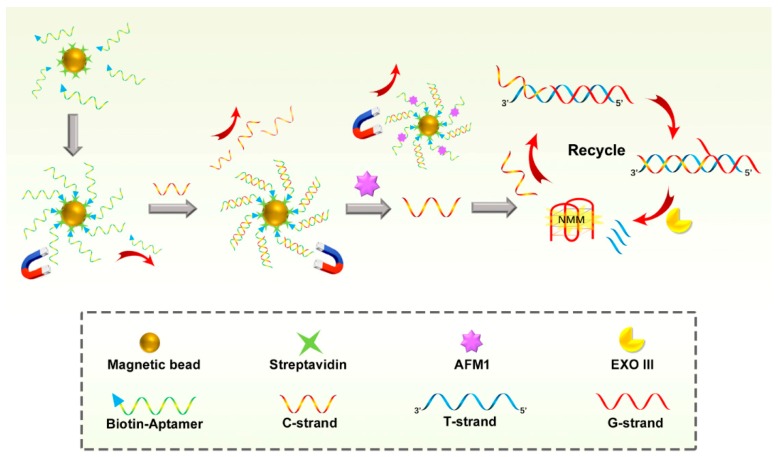
Schematic representation of sensitive fluorescent detection of AFM1 based on Exo III-assistant signal amplification recycles.

**Figure 2 nanomaterials-09-00104-f002:**
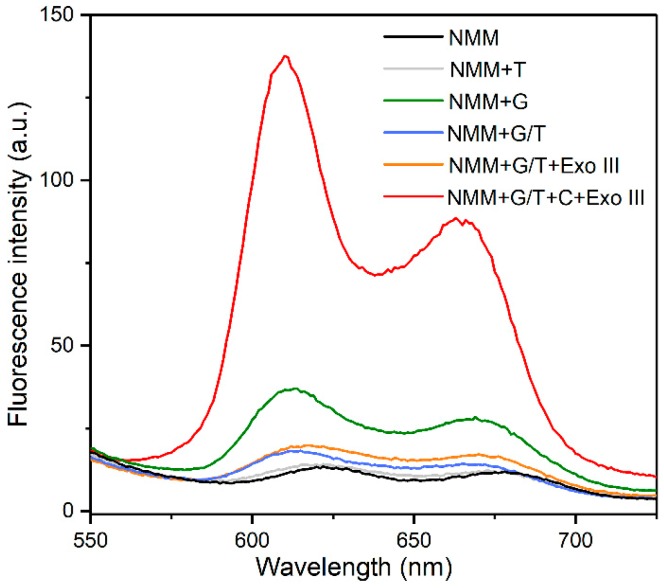
G-quadruplex formed by G-strand enhanced the fluorescence intensity and the amplification recycle activated by Exo III generate significant increase in the fluorescence intensity.

**Figure 3 nanomaterials-09-00104-f003:**
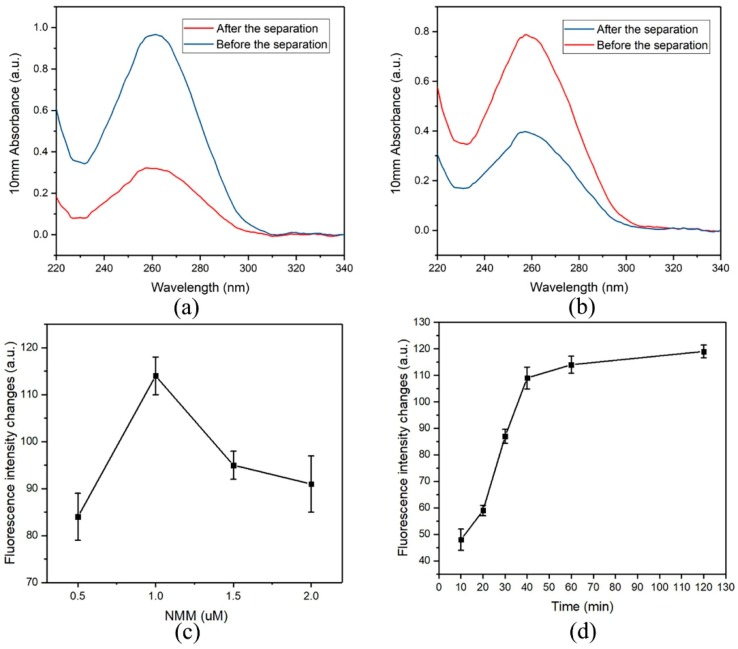
(**a**) UV-vis of the aptamer before (5 µM) and after (1.6 µM) the separation step; (**b**) UV-vis of C-strand DNA before (4.7 µM) and after (2.3 µM) the capture step; (**c**) The concentration of NMM in the signal amplification recycle; (**d**) The reaction time of the signal amplification recycles.

**Figure 4 nanomaterials-09-00104-f004:**
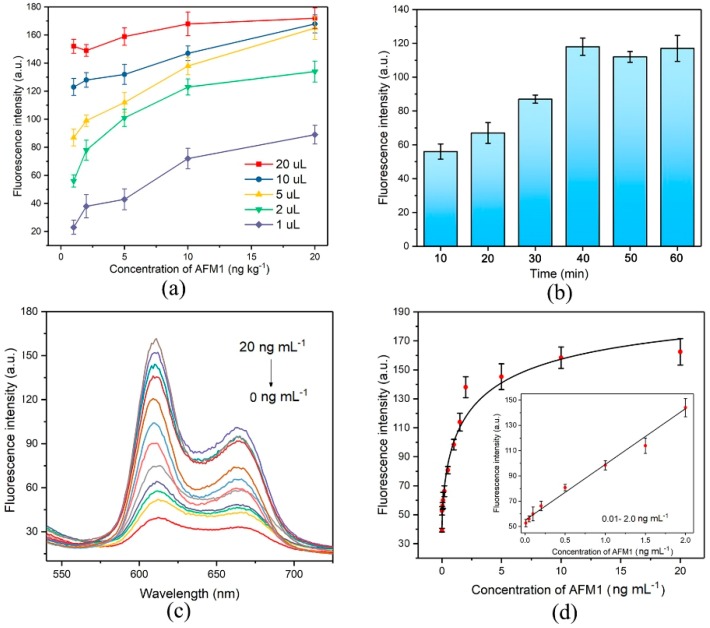
The optimization of the experiment: (**a**) Different amount of first-step resultant solution was submitted to the Exo III-assisted recycling amplification in the presence of AFM1. (*n* = 3); (**b**) The AFM1 activated first-step displacement time; (**c**) The fluorescence spectra changes according to the concentration of AFM1 (from top to bottom: 20 ng mL^−1^ to 0 ng mL^−1^); (**d**) The calibration curve was achieved under optimized conditions.

**Figure 5 nanomaterials-09-00104-f005:**
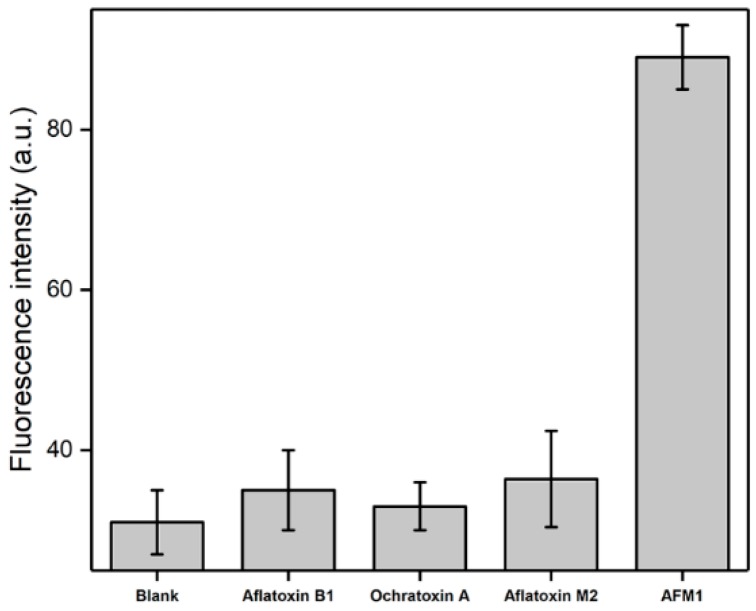
Selectivity evaluation of the developed aptasensor for the detection of AFM1 (1 ng mL^−1^) against other control mycotoxins of AFB1, AFM2 and Ochratoxin A at the same concentration.

**Figure 6 nanomaterials-09-00104-f006:**
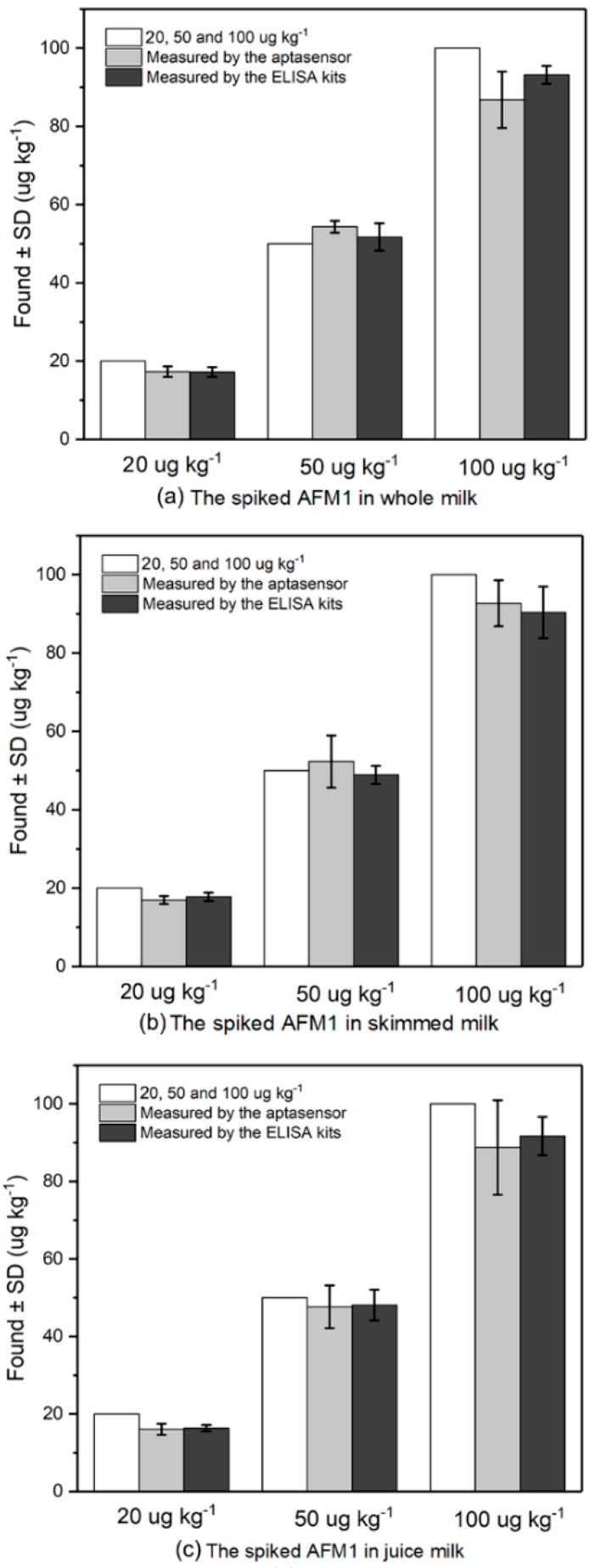
Additive study by adding different concentrations of AFM1 (20, 50 and 100 µg kg^−1^) in (**a**) whole milk, (**b**) skimmed milk and (**c**) juice milk. The obtained results by the developed aptasensor and commercial ELISA kits are compared (in white) (*n* = 3).

**Table 1 nanomaterials-09-00104-t001:** Applications of the established sensor for detection of AFM1 in different milk samples and the verification by commercial ELISA kit. (*n* = 3).

Samples	Added (ng kg^−1^)	Aptasensor	ELISA Kit
		Recovery (%)	Recovery (%)
Whole milk	20	86.47 ± 6.67	86.12 ± 6.23
	50	108.67 ± 3.03	103.40 ± 7.05
	100	86.80 ± 7.18	93.20 ± 2.30
Skimmed milk	20	84.67 ± 5.11	88.91 ± 5.37
	50	104.67 ± 13.32	97.86 ± 4.58
	100	92.70 ± 5.88	90.41 ± 6.60
Juice milk	20	80.13 ± 2.12	81.81 ± 3.05
	50	95.33 ± 11.02	96.18 ± 7.93
	100	86.50 ± 6.21	91.71 ± 4.94

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
