# Peer review of "Turn-On Fluorescence Aptasensor on Magnetic Nanobeads for Aflatoxin M1 Detection Based on an Exonuclease III-Assisted Signal Amplification Strategy"

_nanomaterials, 2019, doi:10.3390/nano9010104_

Reviewer 1 Report

This paper describe a methodology for the detection of Aflotoxin M1 in milk. This is a well structured paper and the results apparently are scientifically relevant. 

Author Response

Thank you very much for your kind comments.

Reviewer 2 Report

In this paper the authors present the development of an aptasensor for the label free detection of AFM1.Generally, the paper is well written. Clear English is used. The introduction gives appropriate references and explanations of the state of the art. Only lacks of the proper explanation of T-G-strands in the aptasensor developed. In the experimental section, I would include the table S1 that is in supplementary as long as the sequences are essential to understand this development. Results are excellent because the authors developed an aptasensor with very good sensitivity and easy to use. However several points need to be improved.

In particular,

Abstract line 35 (and so on) "detection limit of 0.01 ng mL-1". The authors should use the same units along the text in order to compare with the legal limits for maximum residue level mentioned in the introduction section (ng kg-1).

Introduction line 84: Although the “constructed” aptasensors. Reiteration of the word “constructed”. Instead, the authors may use other words as “developed”.

Introduction line 92 to 96. Sentence too long, difficult to understand. Moreover, this sentence explain the basics of this aptasensor. Therefore, I suggest the authors to rewrite it again in order to make it fully understood. In addition, the authors should explain herein the importance of the T-strand and G-strand.

Introduction line 110-112. Sentence too long and redundant with the next.

Experimental line 140. Why the authors wash with PBS instead of the Tris buffer?

Experimental line 141: in which buffer is diluted the C-strand?

Experimental line 146. Which buffer do the authors use? Do the authors denature the G-strand previously in order to facilitate the hybridization of both strands?

Experimental line 157. “NMM with 1 μM of a final concentration was added to the system and incubated for 15 min”. Change suggestion: “NMM, with a final concentration of 1 μM, was added to the system and incubated for 15 min”

Results and discussion section

Why the NMM G-strand mixture gives only a 3.2 fold increase in the fluorescence intensity? It might be expected to give a similar signal (7.2 fold) as the whole complex with all the sequences and ExoIII, as all the G-strands are available to bind NMM. Please give a reasonable explanation.

Binding aptamers to the magnetic nanobeads: please indicate the number of replicates of that experiment and include standard deviation.

Line 237. Figure 3. “the concertation of NMM in the signal amplification recycle (d) The optimization of the concertation”. Change to “the concentration of NMM in the signal amplification recycle (d) The optimization of the concentration”.

Line 243. Why the authors use the G/T duplex instead of the G-strand alone to optimize the NMM concentration?

Analysis of AFM1 in spiked milk samples

-Due to the excellent results obtained in buffer, I would have expected a similar calibration curve in milk samples in order to calculate the Limit of Detection as well as the working range of the aptasensor in real samples. Otherwise, the paper is weak in real samples analysis.

-Additionally, the results given in table 1 should be exposed in graphs comparing each sample with the ELISA kit. Statistical analysis should also be included showing the deviation from the elisa kit (if any). In addition, the authors do not mention the number of replicates used in this analysis.

Conclusions

Line 324: “The magnetic nanobeads were innovatively introduced”. Actually, there are too many examples in which magnetic beads are used in biosensing and, therefore, it is not a real innovation.

Line 326. From my point of view, this sentence is not the best to explain the excellent results obtained in this work. The detection of AFM1 is simply indirect, there is no more words to describe it.

Line 327. If the authors affirm, “signal enhancement was highly dependent on the concentration of the target AFM1” then they should give the correlation covariance data on every calibration curve performed in order to give real facts on it.

Line 328. “aptasensors were sensitive to AFM1 low to approximately 0.01 ng mL-1”. The authors should calculate the limit of detection of the aptasensor both, in buffer and milk samples.

Line 331. The universality of this platform is unclear given that all the aptasensors rely on the affinity of the aptamer by its target (which has to be higher than that for the complementary strand, otherwise no displacement will be detected).

Author Response

Thank your valuable comments on our manuscript entitled “Turn-on fluorescence aptasensor on magnetic nanobeads for aflatoxin M1 detection based on an Exonuclease III-assisted signal amplification strategy”. We have studied these comments carefully and have made point-to-point corrections according to each comment as listed in below section "#Responses to Reviewer 2". We have provided the revise manuscript and revised supporting information. The main corrections are highlighted in red. We believe the quality of this revised manuscript have been improved significantly with your help. Thanks for your consideration and we look forward to hearing from you in due course.

# Responses to Reviewer 2:

Abstract line 35 (and so on) "detection limit of 0.01 ng mL-1". The authors should use the same units along the text in order to compare with the legal limits for maximum residue level mentioned in the introduction section (ng kg-1).

Response:

Special thanks for your comments on our paper and we have revised it according to your comments:

The density of milk is 1.0288 g mL-1, so 1 mL of milk is about 1.0288 g, and 0.01 ng mL-1 equivalent to 9.73 ng kg-1. We added this information in the manuscript. Line 35 and 289.

Introduction line 84: Although the “constructed” aptasensors. Reiteration of the word “constructed”. Instead, the authors may use other words as “developed”.

Response:

Thank you for pointing this out.

We revised as “developed” according to our comment in this sentence. Line 83.

Introduction line 92 to 96. Sentence too long, difficult to understand. Moreover, this sentence explain the basics of this aptasensor. Therefore, I suggest the authors to rewrite it again in order to make it fully understood. In addition, the authors should explain herein the importance of the T-strand and G-strand.

Response:

Thank you for pointing this out.

We rewrite this sentence and added a sentence to explain the importance of the T-strand and G-strand in the manuscript. Line 91-96.

Introduction line 110-112. Sentence too long and redundant with the next.

Response:

We revised this sentence in the manuscript. Line 110-112.

Experimental line 140. Why the authors wash with PBS instead of the Tris buffer?

Response:

We thank you for pointing out this mistake.

We are very sorry to realize that it is a clerical error, and it should be Tris buffer in this section. We used Tris buffer in this experiment. We revised it in Line 141.

Experimental line 141: in which buffer is diluted the C-strand?

Response:

We used Tris buffer to dilute the strands in this experiment. We added this information in Line 142.

Experimental line 146. Which buffer do the authors use? Do the authors denature the G-strand previously in order to facilitate the hybridization of both strands?

 Response:

We used Tris buffer in this experiment. The strands were pre-heated to 90°C for 3 min, followed by cooling slowly to 37°C for 1 h of incubation to facilitate the hybridization of both strands. We added this information in this manuscript. Line 148-149.

Experimental line 157. “NMM with 1 μM of a final concentration was added to the system and incubated for 15 min”. Change suggestion: “NMM, with a final concentration of 1 μM, was added to the system and incubated for 15 min”

Response:

Thank you for your comments on this issue.

We revised this sentence according to your comments in the manuscript. Line 159.

Why the NMM G-strand mixture gives only a 3.2-fold increase in the fluorescence intensity? It might be expected to give a similar signal (7.2 fold) as the whole complex with all the sequences and ExoIII, as all the G-strands are available to bind NMM. Please give a reasonable explanation.

Response:

Thank you for pointing this out.

When the C-strand DNA was added to the whole complex with all the sequences and ExoIII, the C-strand hybridized with the T-strand to form the 3’-blunt terminus on the duplex DNA. Subsequently, it was digested by Exo III, releasing the G-stand to fold into the quadruplex structure. The C-strand was able to trigger the signal amplification recycle, and Exo III then catalyzes the stepwise removal of mononucleotides from this terminus, releasing the C-strand DNA for initiating another cycle and ultimately liberating the quadruplex-forming oligomer to form a quadruplex structure. Thus, a remarkable increase in fluorescence intensity (7.2-fold) was observed, and it was much stronger than that caused by the G-strand alone (3.2-fold). We rewrite this section in Line 215-226.

Binding aptamers to the magnetic nanobeads: please indicate the number of replicates of that experiment and include standard deviation.

Response:

Thank you for pointing this out.

We added this information in this manuscript. Line 232 and 234.

Line 237. Figure 3. “the concertation of NMM in the signal amplification recycle (d) The optimization of the concertation”. Change to “the concentration of NMM in the signal amplification recycle (d) The optimization of the concentration”.

Response:

Thank you for pointing this out.

We revised this sentence according to your comments in the manuscript. Line 239.

Line 243. Why the authors use the G/T duplex instead of the G-strand alone to optimize the NMM concentration?

Response:

Thank you for pointing out this mistake.

We apologize for this clerical error in our manuscript, and it should be G-strand DNA instead of G/T duplex. As we mentioned in Line 208-210, the mixture of NMM and G/T duplex DNA exhibited an insignificant change in fluorescence intensity because the formation of G/T duplex led to the lack of guanine-rich sequence (G-strand).  We revised it in Line 246.

Analysis of AFM1 in spiked milk samples

-Due to the excellent results obtained in buffer, I would have expected a similar calibration curve in milk samples in order to calculate the Limit of Detection as well as the working range of the aptasensor in real samples. Otherwise, the paper is weak in real samples analysis.

Response:

Thank you for pointing this out.

In this study. The feasibility of the sensor was verified by introducing different concentrations of AFM1 in Tris buffer (Figure 4a). Considering the practicability of the sensor, like commercial ELISA kits, we established the calibration curve by spiking AFM1 in the negative control milk samples which were pre-treated according to the sample treatment method. So, the mentioned detection limit and working range in this manuscript was based on the real sample, and this is why we directly use the calibration curve for sample detection. We added this in formation in Line 277-279.

-Additionally, the results given in table 1 should be exposed in graphs comparing each sample with the ELISA kit. Statistical analysis should also be included showing the deviation from the elisa kit (if any). In addition, the authors do not mention the number of replicates used in this analysis.

Response:

Thank you for your meaningful comments.

We added a figure (Figure 6) in the manuscript to compare the aptasensor’s performance with ELISA kits. We also added the standard deviation data in Table 1. The number of replicates used in this analysis was also added in the captions of the Figure 6 and Table 1.

Conclusions

Line 324: “The magnetic nanobeads were innovatively introduced”. Actually, there are too many examples in which magnetic beads are used in biosensing and, therefore, it is not a real innovation.

Line 326. From my point of view, this sentence is not the best to explain the excellent results obtained in this work. The detection of AFM1 is simply indirect, there is no more words to describe it.

Response:

Your meaningful comments are very much appreciated.

We realized that it is not appropriate to describe the introduction of magnetic nanobeads using “innovatively”. We removed this word in this sentence. We also re-organized the Conclusion section in this manuscript to point out the advantages of the developed sensor. Line 332-343.

Line 327. If the authors affirm, “signal enhancement was highly dependent on the concentration of the target AFM1” then they should give the correlation covariance data on every calibration curve performed in order to give real facts on it.

Response:

Thank you for your meaningful comments.

We calculated the correlation coefficient on the calibration curve in the working range. The correlation coefficient calculated by Excel was 0.995, which showed that the signal enhancement was dependent on the concentration of the target AFM1 in the working range. We added the correlation coefficient in Line 287.

Line 328. “aptasensors were sensitive to AFM1 low to approximately 0.01 ng mL-1”. The authors should calculate the limit of detection of the aptasensor both, in buffer and milk samples.

Thank you for pointing this out.

In this study. The feasibility of the sensor was verified by introducing different concentrations of AFM1 in Tris buffer (Figure 4a). Considering the practicability of the sensor, like commercial ELISA kits, we established the calibration curve by spiking AFM1 in the negative control milk samples which were pre-treated according to the sample treatment method. So, the detection limit was based on the real samples, which is more meaningful for real application. We added the information in Line 337.

Line 331. The universality of this platform is unclear given that all the aptasensors rely on the affinity of the aptamer by its target (which has to be higher than that for the complementary strand, otherwise no displacement will be detected).

Response:

Thanks for your professional comments.

We realized that our description is inaccurate and misleading. If this this sensor used for other targets, the optimization of aptamer/complementary strand combination is important. We re-organized the Conclusion section in this manuscript. Line 341-343.

Once again, we thank for your persuasive comments on our manuscript.

Reviewer 3 Report

Comments to the paper nanomaterials-419793

In this manuscript, the authors describe the conjugation of magnetic nanoparticles with different DNA strands in order to detect aflatoxin M1 (AFM1). Although all procedures used such as, streptavidin-biotin combination, the aptamer-AFM1 selectivity, or NNM-G-quadruplex fluorescence enhance, are already well known. But certainly, the combination of all of them result in a highly sensitive sensor.

The experiments propose address the main goal of the paper, although I have one concern, the sensor is concentration dependent with many elements of the sensor, NNM, ExoIII, aptamer, C-strand, G-strand, that authors resolve by saturation to maximum signal all parameters. Only, discussion about ExoIII and AFM1 concentration are shown (Fig 4). It may be interesting to define other concentration species. For all this I consider the manuscript interesting and suitable to be published in Nanomaterials

Author Response

Thank you for your kind comments on our manuscript entitled “Turn-on fluorescence aptasensor on magnetic nanobeads for aflatoxin M1 detection based on an Exonuclease III-assisted signal amplification strategy”. We have made point-to-point corrections for this manuscript. The main corrections are in red and the responses to the comments are as following.

We believe the quality of this revised manuscript have been improved significantly with your help. Thanks for your consideration and we look forward to hearing from you in due course.

 Responses to Reviewer 3:

The experiments propose address the main goal of the paper, although I have one concern, the sensor is concentration dependent with many elements of the sensor, NNM, ExoIII, aptamer, C-strand, G-strand, that authors resolve by saturation to maximum signal all parameters. Only, discussion about ExoIII and AFM1 concentration are shown (Fig 4). It may be interesting to define other concentration species. For all this I consider the manuscript interesting and suitable to be published in Nanomaterials.

Response:

Thank you very much for your comments.

We also optimized the amount of the C-strand (first-step resultant solution) which was used in the amplification recycles to produce enough strong fluorescence signal (Fig 4a). As shown in Fig.3d, based on the optimized conditions, the enhanced signal levelled off after a 40 min of incubation, and the produced fluorescence signal was enough for detection usage, and it indicated that the applied G-strand concentration fulfill the application need in the sensor.

Once again, we thank for your persuasive comments on our manuscript.

Round  2

Reviewer 2 Report

All issues have been positively addressed

Author Response

Thank you for your comments which help to improve the quality of this manuscript.